# Patient, family member, and health care provider perspective on barriers and facilitators to diabetic retinopathy screening in Thailand: A qualitative study

**Geetha Kumar[1]☉, Saranya Velu[1]☉, Shahina Pardhan[2], Raju Sapkota[2], Paisan Ruamviboonsuk[3], Mongkol Tadarati[3], Peranut Chotcomwongse[3], Variya Nganthavee[3], Warisara Pattanapongpaiboon[3], Rajiv Raman[1] ***

**1** Shri Bhagwan Mahavir Vitreoretinal Services, Sankara Nethralaya, Chennai, Tamil Nadu, India, **2** Vision and Eye Research Institute, School of Medicine, Anglia Ruskin University, Cambridge, United Kingdom, **3** Department of Ophthalmology, Rajavithi Hospital, Bangkok, Thailand

☉ These authors contributed equally to this work.
* rajivpgraman@gmail.com

## Abstract

### Objectives

Diabetic retinopathy (DR) can cause significant visual impairment which can be largely avoided by early detection through proper screening and treatment. People with DR face a number of challenges from early detection to treatment. The aim of this study was to investigate factors that influence DR screening in Thailand and to identify barriers to follow-up compliance from patient, family member, and health care provider (HCP) perspectives.

### Methods

A total of 15 focus group discussions (FGDs) were held, each with five to twelve participants. There were three distinct stakeholders: diabetic patients (n = 47) presenting to a diabetic retinopathy clinic in Thailand, their family members (n = 41), and health care providers (n = 34). All focus group conversations were transcribed verbatim. Thematic analysis was used to examine textual material.

### Results

Different themes emerged from the FGD on knowledge about diabetes, self-care behaviors of diabetes mellitus (DM), awareness about DR, barriers to DR screening, and the suggested solutions to address those barriers. Data showed lower knowledge and awareness about diabetes and DR in both patients and family members. Long waiting times, financial issues, and lack of a person to accompany appointments were identified as the major deterrents for attending DR screening. Family support for patients was found to vary widely, with some patients reporting to have received adequate support while others reported having received minimal support. Even though insurance covered the cost of attending diabetes/ DR screening program, some patients did not show up for their appointments.

**Data Availability Statement:** All relevant data are within the paper and its Supporting Information files.

**Funding:** This study was supported by Lions Club International Foundation. SFP2050/UND. The funders had no role in study design, data collection and analysis, decision to publish, or preparation of the manuscript.

**Competing interests:** The authors have declared that no competing interests exist.

## Conclusion

Patients need to be well-informed about the asymptomatic nature of diabetes and DR. Communication at the patient level and shared decision-making with HCPs are essential. Family members and non-physician clinicians (such as diabetes nurses, diabetes educators, physician assistants) who work in the field of diabetes play a vital role in encouraging patients to attend diabetes and DR follow-ups visits regularly.

## Introduction

Diabetes mellitus (DM) is a global public health problem. In 2019 it was estimated that there were 463 million people living with diabetes, and the number is projected to rise to 578 million by 2030 [1]. In Thailand, the prevalence of diabetes increased from 6.9% to 8.9% from 2009 to 2014 [2] and is expected to rise to 9.8% by 2030 [3]. Although a large number of diabetic patients are correctly identified and treated in Thailand, poor control of blood glucose levels has been reported [4].

Patients who are unable to control their blood glucose levels are more likely to develop major complications including diabetic retinopathy (DR) [5]. DR is a condition that damages blood vessels in the retina. These blood vessels can leak, causing vision loss or even blindness [6]. Nearly 1.4% of blindness in Southeast Asia (including Thailand) is due to DR [7], and 24–31% of people with type 2 diabetes have DR (21–23% with non-proliferative DR and 2.3–9.4% with a proliferative DR, which is a more severe form) [8–11]. According to Euswas N et al, the prevalence of DR in the Thai population ranges from 5.0 to 6.9% [12].

Blindness due to DR can be reduced by proper screening and timely treatment of DR [13], and by adopting healthy lifestyle modifications [14]. Qualitative studies from the United States [15], Canada [16], the United Kingdom [17], and Africa [18] using focus group discussions have identified several barriers to DR screening. These pertained to poor knowledge and awareness about DR [19], lack of skilled manpower and facilities for performing DR examination [20], inadequate interaction between hospital staff and patients, non-integration of DR screening with other health screening programs during the same visit, ethnic diversity [17], fear of having dilated fundus examination, religious beliefs that divine justice will prevail [18], and delay in decision-making to attend DR screening [21].

A recent community-based study from Thailand reported that less than 50% of diabetic patients underwent annual DR screening [6]. However, there are limited data pertaining to the barriers to DR screening and its uptake in Thailand. The purpose of this study was to identify barriers to attending follow-up appointments for DR examination from the perspective of the patients, health care providers (HCPs), and family members, and to identify factors that influence patient compliance with attending DR screening and follow-up visits.

## Methods

This descriptive qualitative study was conducted between August and December 2019. Focus group discussions (FGDs) were performed in five DR outpatient clinics by three authors (VN, WP, and MT who were trained in qualitative research methods by PC). Each hospital examines patients from the five different regions of Thailand (north, northeast, west, central, and south). The study was approved by the institutional review board, the Ethics Committee, Rajavithi Hospital (approval no: 076/2561), and adhered to the tenets of the declaration of Helsinki.

## Inclusivity in global research

Additional information regarding the ethical, cultural, and scientific considerations specific to inclusivity in global research is included in the S1 Text.

## Sampling

Using the maximum variation sampling, a minimum of five to a maximum of 12 participants were recruited in each group [22, 23]. The stakeholders included patients, family members, and health care providers. There were 3 FGDs from each outpatient clinic, making a total of 15 FGDs. The selection of a varied group of participants enabled us to obtain diverse perceptions on the study topic which was to understand, perceptions and awareness about DR, the need for regular eye screening, perceived barriers in seeking eye care, in addition to awareness about diabetes and diabetic-related complications.

## Data collection

A focus group agenda (S1 Text) was designed based on the literature [24] and prior interactions with patients, family members, and HCPs. Around thirty-four HCPs from various departments were recruited who had 5 or more years of experience working with diabetic patients. Prior to the FGD, all participants were briefed about the study. Written informed consent was obtained from every participant. The information gathered from the participants included an understanding of diabetes, their experiences of living with diabetes, lifestyle changes, care-seeking behaviors, understanding of barriers to DR and the importance of attending DR screening, and possible suggestions for improving compliance with DR screening.

Patients were invited to take part while attending the DR screening at the hospital. Patients were invited to participate if they were 18 years or older, had type 1 or 2 DM, and were excluded if they had no perception of light in either eye, were unwell to participate, or had a physical or mental disability. Family members who accompanied patients to their hospital appointment were invited to take part, forming the second stakeholder group.

Each group discussion session lasted 45–60 minutes. The data were collected in the Thai language. The entire conversation was audio-recorded and transcribed verbatim, translated into English by the general practitioners, and double-checked. The transcribed data were re-read several times, and the original recordings were listened to several times to ensure the accuracy of the transcription. Several issues/themes emerged from the focus group discussion, including management of diabetes, DR awareness, and DR screening compliance. To protect participant confidentiality, all data was anonymized.

Of the total 122 participants, 27.04% were males. The mean age and standard deviation (SD) for the participants in the individual focus group are provided in Table 1. The mean duration of diabetes was 9.23 (±9.49) years (self-reported).

## Data analysis

Data analysis followed the framework analytical approach [25]. Management and interpretation of qualitative data following an iterative data handling process was undertaken by gaining familiarity with each of the transcripts through repeated readings [26]. We carried out a systematic method of organizing our data into spreadsheets, keeping in mind our research objectives, and listed out several categories such as 'knowledge about diabetes, 'behaviors to DM care', 'awareness about DR care', and 'barriers to DR screening'. We then extracted relevant text from each FGD related to these categories and went through a process of indexing or

**Table 1. Characteristics of patients, family members and health care givers.**

| Characteristics of Patients | n(%) | Characteristics of Family members | n(%) |
|---|---|---|---|
| **Gender** | | **Gender** | |
| Male | 15 (31.91) | Male | 9 (22) |
| Female | 32 (68.05) | Female | 32(78) |
| **Age** | 57.61 ±12.95 | **Age** | 49.60 ± 13.84 |
| **Duration of diabetes in years** | 9.23 ± 9.49 | | |
| **Occupation** | | **Occupation** | |
| Businessman | 1 (2.12) | Accountant | 1 (2.43) |
| Driver | 1 (2.12) | Businessman | 4 (9.75) |
| Employee | 6 (12.76) | Civil servant | 2 (4.87) |
| Factory worker | 1 (2.12) | Farmer | 15 (36.58) |
| Farmer | 6 (12.76) | General contractor | 2 (4.87) |
| General worker | 4 (8.51) | General employee | 1 (2.43) |
| Housekeeping | 8 (17.02) | House Keeper | 2 (4.87) |
| Housemaid | 4 (8.51) | Housekeeping | 4 (9.75) |
| Retired | 5 (10.63) | None | 2 (4.87) |
| Routine work | 1 (2.12) | Office work | 1 (2.43) |
| Salesman | 8 (17.02) | Public Servant | 1 (2.43) |
| Security officer | 1 (2.12) | Retired | 1 (2.43) |
| Welder | 1 (2.12) | Salesmen | 5 (12.19) |
| **Characteristics of Health care providers** | **n(%)** | | |
| Gender | | | |
| Male | 9 (26.47) | | |
| Female | 25 (73.52) | | |
| Age | 41.55±9.45 | | |
| Year of experience | 17.10±11.16 | | |
| Occupation | | | |
| Nurse | 9 (26.47) | | |
| Nurse aid | 8(23.52) | | |
| Nutritionist | 1 (2.94) | | |
| Ophthalmologist | 2(5.88) | | |
| Pharmacist | 2(5.88) | | |
| Physical therapist | 1 (2.94) | | |
| Registered Nurse | 6(17.64) | | |
| Registrator | 1 (2.94) | | |
| Technician | 1 (2.94) | | |
| Traditional Medicine Doctor | 1 (2.94) | | |
| Laboratory scientist | 1 (2.94) | | |
| Clerk | 1 (2.94) | | |

sifting through the data; sorting and selecting quotes and placing them under the appropriate categories. By comparing and contrasting the data and identifying the categories that might be usefully merged and those that stood alone, we were able to refine our categories and lay the groundwork for theme creation. In developing themes, we looked for patterns and made decisions on what specific themes would best explain our data and provided important insights.

## Results

Various themes that best explained the data and addressed our research questions were knowledge of DM, behaviors of DM care, awareness about diabetic retinopathy, barriers to DR screening, and suggestions to improve DR screening attendance. The findings are discussed below, along with the supporting quotes.

### Knowledge of diabetes

The majority of the patients were diagnosed after they experienced symptoms including exhaustion, frequent urination, foot numbness, leg pain, and weight loss. Minimal patients reported being diagnosed with DM at a regular annual health examination. Four participants were incidentally diagnosed with diabetes when they attended eye check-ups for cataracts. The primary sources from which patients reported receiving information about diabetes included volunteers, doctors, nurses, and posters.

> *"I urinated every hour. I went to the hospital and the doctor said that my 72-kg body weight was obese, and my blood sugar was high. The doctor told me to lose 10 kilograms of weight. After I had lost 10 kgs, I went back to see the doctor and he told me that I had diabetes. (FGD 2)"*

> *"I was a truck driver. I came to the hospital for eye surgery, but my blood pressure was very high, so I could not have surgery. After having blood tests, I was diagnosed with hypertension and diabetes. I have never received a check-up before, but my vision was poor, so I first came to the hospital to have an eye examination. (FGD 1)"*

Seven patients were worried about diabetic symptoms such as polyuria, fatigue, tiredness, and sleep disorder. Others accepted the inevitability of acquiring the disease. One patient was distressed after learning she was diabetic, but she eventually accepted after learning that diabetes is a common disease. A small number of patients compared diabetes with another disease such as cancer or having an accident but only one patient reported to being aware of the fact that diabetes can affect the eyes and kidneys. Patients whose blood sugar levels did not improve even after taking medication appeared unwilling to return for the follow-up care, making them seek treatment at another hospital.

> *"At first, I was upset but then I realized that many other people also have diabetes too, and that made me feel better. (FGD 1)"*

> *"I have been diagnosed with diabetes for 4 years. I also had frequent urination and itching vagina. I also feel fatigue. My blood sugar was over 200. I went to see a doctor at private hospital but the medications there did not lower my blood sugar, so I changed to this hospital. The doctor gave me NPH here and it worked well with me. I used to have a Mixtard but it didn't work well. (FGD 3)"*

Family members reported feelings of sympathy, empathy, frustration, and worry for diabetic patients. Many family members also echoed fear of death, blindness, and the possibility of diabetic patient's legs being amputated. They reported concerns around delayed wound healing, impaired vision, and joint stiffness.

> *"A known diabetic patient passed away who lived far away from the hospital. Getting him to the hospital was difficult, we sympathise for him. His excessive blood sugar level caused numbness in his hands and feet. (FGD 3)"*

*"My dad had history of diabetes and he was very cautious to avoid wounds. He had an ulcer in leg, then he passed away due to diabetes (FGD 5)."*

## Behaviors of DM care

The majority of patients preferred changing their eating habits and adopting a healthy lifestyle to control their diabetes rather than depending on only medication. The doctor's advice appeared to have substantially boosted the patients' attitude towards a positive state of mind during the consultation. While some participants reported maintaining healthy blood glucose levels by a combination of medication and adhering to a strict diet as suggested by the doctor, others reported controlling their blood sugar levels by solely drinking plenty of water. One patient seemed determined about maintaining her health through strict dietary restrictions over the fear of developing DR and neuropathy.

*"The doctor said I nearly had diabetes and asked me to follow a controlled diet and come for the follow up for 1 month. If the blood sugar was still rising, I would need to take medications. I had tried diet control. (FGD 1)"*

*"I was diagnosed with diabetes, but I ignored it for two years until I had numbness. I didn't come (to hospital) because I had no symptoms. (FGD 3)"*

*"I try to drink water only since I realize that uncontrolled blood sugar would lead to many illnesses and my relatives would be affected too. I also realize that diabetes can lead to shock, so I regularly take medications and restrict my diet. (FGD 2)"*

Most of the family members reported actively supporting and caring for patients with diabetes and shared responsibilities of disease management including providing blood sugar measuring devices, reminding patients of appointments, providing social and emotional support to cope with diabetes, and helping to change to a healthy diet. They remind patients about issues such as lifestyle changes and adhering to medication regimes as instructed by the doctor. While a few family members had no trouble monitoring the patient's health because the patients themselves were diligent in keeping on track by eating well, taking medications on time, and following the doctor's advice, there were others who had difficulty dealing with patients who were stubborn towards following the advice from the doctors.

*"I am very confident because I take care of her by myself. Now she doesn't have diabetes. (FGD 1)"*

*"My mother feels like she knew everything. She always comes to visit clinic, but she sometimes takes medication. Her blood sugar varies from 170 to 240. After buying her a blood sugar measuring machine, she increased her self-awareness. I tried to prepare medication for her, but she refuses. She eats a lot of rice and boiled vegetable. She refuses to take herbal drugs. (FGD 4)"*

HCPs reported that there were two categories of patients: 1) Those who understood the necessity of diabetic treatment and who therefore attended follow-up appointments for diabetes on a regular basis, and 2) patients who did not understand or were uninformed of the importance of diabetic treatment and blamed themselves for not benefiting from follow-up appointments. Therefore physician and non-physician clinicians felt that patients should be given a guidebook, posters, books, videos, or brochures with the necessary information

reminding them about the possible complications of uncontrolled diabetes in eyes, feet, etc. Illiterate patients may benefit from being frequently reminded of the complications verbally or through diagrams/photos/videos.

## Awareness about DR

There was lower awareness around DR, for example, participants did not know the difference between DR and glaucoma. One participant thought that DR would always reduce their vision, reporting that DR is present only when the vision is blurred and does not improve after wiping the spectacles. The majority of patients were aware that DR is linked to abnormal blood sugar levels, but only a few knew that DR could lead to vision impairment. Only two participants appeared to be aware of laser treatment and injections for DR management. Interestingly, one patient stated that she will do internet research on DR and then share her knowledge with other diabetic patients to help them understand that "*uncontrolled blood sugar might lead to blindness*". Two patients expressed gratitude for the DR awareness movies that were played during their visits to the hospital. Patients were found to have differing perspectives on surgical treatment by ophthalmologists. Patient compliance was found to be influenced by the level of trust in doctors. For example, one participant reported to being apprehensive about signing the consent form which stated "*ophthalmologists are not responsible for any blindness or complications*", which was the message focused on the risks that were raised, such as the type and effects of the treatment, the results of the surgery, or the patient concerns around the disease prognosis.

> "*The doctor told me that I had glaucoma, and I am wondering if the glaucoma and diabetic retinopathy is the same entity or not? Do I have a chance to develop diabetic retinopathy? I believe that blood sugar depends on the food we eat. (FGD 1)*"

> "*I study about diabetic retinopathy on YouTube by myself. I also share the information with other diabetic patients and realize that uncontrolled blood sugar would lead to blindness. (FGD 2)*"

> "*Before going to the surgery, you have to sign for the informed consent which stated, "The ophthalmologists are not responsible for any blindness or complications if occur". I feel frightened since there is no responsible person to take care. (FGD 4)*"

Awareness of DR was found to be limited amongst the family members. Very few family members knew how DR is treated, and knew little about drugs, injections, lasers, and surgeries for treating DR. Some family members reported that diabetes is linked to increased intraocular pressure. Only four family members stated that damaged blood vessels due to diabetes cause blurry vision and can lead to blindness.

> "*High blood sugar can affect blood vessels leading to blindness. Annual screening is required. I know only sugar control is necessary. (FGD 2)*"

According to HCPs, a large majority of the patients appeared to be aware of the need for diet control and taking medicine. However, HCPs also reported that patients who had high blood sugar (glycated hemoglobin) levels seemed to adopt strict control of diet only prior to attending hospital appointments. HCPs reported that patients were often tempted to eat unhealthy foods resulting in poor blood sugar control. HCPs reported that patients were not always aware of what to do in an emergency situation (hypo/hyper glycaemic condition) and sometimes would wait for their next appointment despite the need to attend sooner.

Moreover, patients tend to use traditional and self-prescribed herbal items to lower blood sugar levels.

*"Some patients also lied to us. Sometimes they use to say that they don't eat any rice or food at all time, but actually they would have had three bananas, one piece of bread, and sweet drinks and some even had the temptation to eat more dessert items. (FGD 1)"*

According to HCPs, some patients did not appear to take appropriate steps to lower their blood glucose levels even when they were aware of the normal blood glucose range. HCPs noticed that patients would seek treatment for diabetes as soon as they become aware of their condition because they were concerned it would result in visual loss. HCPs also reported that the majority of participants were anxious about the course of recovery of diabetes or when they were advised to see an ophthalmologist for their eyes.

*"Patients know that they should continuously take oral medicine. They understand that diet control is the best method to control blood sugar. Some can't control blood sugar because they just want to eat. Some, after having good sugar levels, stopped taking medicine on their own. Then, they came back to the hospital with diabetes ketoacidosis. (FGD 2)"*

## Barriers to DR screening

Barriers to DR screening included a lack of availability of ophthalmologists, long waiting times, financial constraints, transportation problems, the absence of accompanying individuals to attend appointments, and ignorance of DR screening. These challenges were cited by patients as reasons for not going to the hospital. Other reasons included long travel time and forgetting the appointment date.

*"Sometimes I went there in the morning and have to wait until the afternoon. (FGD 2)"*

*"After I just had a laser, both eyes could not see anything. I need people to accompany me [to appointment]. Since I have to come so often [to attend appointments], no one comes with me anymore. (FGD 4)"*

Few family members reported that the diabetic patients were able to take care of themselves by complying with the doctor's suggestions. Few family members reported experiencing financial difficulties since they had to accompany the patients all the time for consultation and for buying medicines. Family members encountered challenges when trying to modify their diets exclusively for diabetic patient.

*"My mother is old so I need to take care of her all time. At least one person needs to stay with her. I am afraid of her having an amputation, so I need to take care of her and take more responsibility to take care of her. (FGD 1)"*

*"A few patients have problems with transportation or a lack of financial support so they decided to follow with their local hospital. (FGD 3)"*

Despite screening services for diabetes being covered by insurance scheme, some patients failed to show up for their appointments, according to the HCPs. Some patients only attended the appointments when their diabetic medicines ran out. Additionally, HCPs indicated that patients did not want to spend time on DR screening; patients did not believe it was important, and the patients would inform the HCPs of their diabetes only if it was getting worse. Financial

problems and lack of family support to accompany patients to appointments were also suggested as deterrents by HCPs for follow-up appointments for DR screening. Anxiety, depression, and family bereavements also played a part.

> *"Many patients have the impression that seeing a doctor is a waste of time. If not seriously ill, they will not come at all. They never take care of themselves nor have a health check-up. (FGD 2)"*

> *"There is an old man whose wife just died recently, he does not come to the hospital afterwards and does not take care of himself anymore. It seems like he gives up. His child tries to bring him here but he denies him. (FGD 4)"*

## Suggestion for DR screening

Most of the patients reported attending follow-up appointments for DR screening if the hospital's reminder system encouraged or alerted them, or when they received a phone call from the doctors reminding them of their appointment. Patients preferred doctors to answer their questions about diabetes management and improving compliance to DR screening. According to some patients, family members had to remind them when an appointment was due. Patients reported that a confirmation phone call about 2 to 3 days before their appointment date would benefit, which would enable them to reschedule any prior appointments when needed.

> *"We have a "Line group" (freeware application software to communicate with many patients) for diabetic patients as well, in which we can change our appointment date within 7 days and also consult for the medications. (FGD 5)"*

Family members suggested that educating people at the village and community levels by holding large diabetes and diabetic eye camps would improve follow-up for DR. Patients who are unable to travel to the hospital due to distance, financial constraints, or other factors should be screened at a primary care center near their home with the help of non-physician clinicians. Providing reminders via telephone calls, posts, e-mails, or SMS, and an exclusive DR screening clinic would avoid long waiting times and enhance patient compliance to attending DR screening. One caregiver identified social stigma, for example, poorer patients felt ignored by the hospital, and suggested that education should be provided by non-physician clinicians (nurse/nurse aid) to poor and elderly patients.

> *"I suggest creating activities and meetings to provide knowledge once or twice a year. I would like health care providers from every department to go to the community. The audience can prevent the disease and can tell other people about the knowledge. We need to prevent it before the treatment. Lots of people without any disease are staying outside the hospital and they don't know about the disease. I would like them to provide knowledge to village and community. In the clinic, there are people to call patients. I told the patients to stick the appointment letter in front of the house. However, I did it and I forgot. (FGD 1)"*

> *"They should separate the diabetic retinopathy clinic from the general eye clinic, so we do not have to wait so long. (FGD 2)"*

HCPs felt that there should be a social worker group to take care of patients who are returning home after attending an appointment with the doctor. Patients should be approached with a positive attitude, and sufficient consultation time should be given to them during every visit (at least 30 minutes). HCPs suggested that coordination with sub-district administrative

organizations (SAO) and village health volunteers (VHV) to enable a group of patients from the same region to travel to the hospital together would be beneficial. Health-care coverage, such as Universal Health Care (UHC), Civil Servant (and public-sector employee) Medical Benefit Schemes (CSMBS), and the Social Security Scheme (SSS) would aid in improving patient follow-up visits. HCPs suggested that patients who miss appointments must be contacted and followed up after a week if they fail to attend, and then if it continues, community hospital staff should contact and encourage them to attend the appointment.

*"There is help from sub-district administration organizations (SAO) and village health volunteers (VHV) to take patients to the hospital. Moreover, we also have a hand-held camera for examining difficult-to-move patients. (FGD 4)"*

*"We have 70 appointments a day. If any of them don't come, I call them and then call again one week later. If they finally don't come, I contact the community hospital to contact patients. Lost follow-up rate is 0.2% because they want to stay in DM clinic. However, hypertension patients lost follow-up rate is 2%. (FGD 5)"*

## Discussion

A number of barriers were identified by the patients, family members, and HCPs. Patient awareness of the need to have regular health checks was found to be very low. Showing videos on diabetes education or presenting patients with diabetic guidebooks would significantly improve awareness about diabetes and its complications. Some patients reported modifying their lifestyle (e.g., eating a healthy diet, exercising) after the diagnosis of diabetes, whilst others didn't adopt this approach. There was less evidence of DM self-care and patients appeared to be keener to leave it up to the doctors to manage their diabetes. Some patients reported being in denial about their diabetes and some would not seek appointments even in an emergency. Patients would rather prefer to receive counseling from a doctor on healthy lifestyle modifications [27]. To facilitate patient understanding about diabetes and DR, communication at the patient level and shared decision-making with HCPs should be encouraged.

Similar to the study by Graham-Rowe et al. [21] and Pardhan et al. [28], our study shows a lack of awareness around DR screening, especially if patients believed their diabetes was under control, or if they were not experiencing any symptoms of vision loss. This emphasizes the need to educate patients about the asymptomatic nature of DR. Some patients did not know the difference between DR and glaucoma, highlighting the need for improving DR knowledge.

Involving families in diabetes care is an important factor in improving outcomes of DM self-care [29]. Family members were found to be distressed by their loved ones having diabetes due to their own limited knowledge about diabetes and its control. Changes in patient health have a significant impact on the whole family, and the family members need to know how to best help diabetic patients [30]. Diabetes management interventions need to involve family members and educate them on how to positively influence healthy behaviors in diabetic patients. Flexible appointment systems can facilitate patient attendance to DR follow-up examinations. Diabetes management interventions should target family members so that they are better equipped to support their diabetic loved one.

Limited understanding of vision loss due to DR was also evident among family members. Family members have a major role in improving DR screening uptake, ranging from accompanying patients to the hospital to supporting the cost of travel, treatment, etc. Inadequate support from family members can hinder access to DR screening [31]. The loss to follow-up for DR translates into a lost opportunity to prevent severe retinopathy and eventual blindness. A

study by Qiu et al. reported having unpleasant feelings during the conversation as a result of not being able to read nonverbal cues such hand gestures, head nods, eye contact, and facial expressions during face-to-face communication [32]. To more fully understand the challenges experienced in receiving diabetic care, future studies should concentrate on individuals who are blind or have severe vision loss.

Non-physician clinicians (often called clinical officers or surgical technicians), play a vital role as in other developing nations which have a shortage of doctors, and may be critical in managing chronic diseases, but they must be well trained in order to gain patients' faith in the quality of care they deliver [33]. All non-physician clinicians working in the field of diabetes care in Thailand are trained in relevant programs, and the quality of care is monitored on a regular basis. Our findings suggest that having a counselor speak with each patient and their family members could have a huge impact on motivating patients to attend DR screening. As diabetes is a multi-faceted disease, its management would benefit from input from various specialties, such as community representatives, physicians, technicians, and nurses, who would share knowledge about recent advances in the self-management of diabetes. Such a unit may be vital in disseminating information to patients about diabetes and DR.

Lack of time, long waiting hours, financial problems, and absence of accompaniment were identified as the major deterrents to follow-up for DR screening, agreeing with previous findings [24]. It is vital that policymakers address these deterrents. Increasing health literacy and spending more time with patients would increase the uptake of DR screening. Schemes such as universal health coverage, the social security scheme, and civil servant medical benefit schemes would help to make screening more accessible. The implementation of Thailand's health policies including the National List of Essential Medicines (NLEM) and Service plan policy covering comprehensive management of DR with a well-established service plan are vital in reducing the burden of blindness due to DR [34].

This study has highlighted important barriers around knowledge of diabetes and DR, and various barriers that may contribute to the high level of DR in Thailand. Our study has some limitations which do not detract in any way from the important findings. The FGD did not include individuals who were too shy to suggest their opinions, and the majority of the participants were female. Due to the lack of a sampling strategy that guarantees coverage of every region in Thailand, the findings may not be generalizable across the whole country due to potential regional differences in socioeconomic position, healthcare services, and beliefs.

## Conclusion

The majority of patients were passive and had great faith in the HCPs. Family participation in DM care can improve health-related behaviors in patients. The involvement of non-physician clinicians working in the field of diabetes has a significant impact on encouraging patients to attend DR screening. Our findings may be useful in designing policies to make DR screening in Thailand more accessible.

## Supporting information

**S1 Text. A focus group agenda.**
(DOCX)

## Acknowledgments

The authors thank Dr. Robin Driscoll for administration support. Dr. Shuba Kumar who is trained in qualitative research conducted a three-day training workshop on qualitative

research methods which was attended by three of the research team (PC, GK and SV). They observed SK conducting interviews and also carried out mock interviews/FGDs using the study guides. Feedback was provided to them to further improve their techniques following which they commenced doing the FGD. We would like to thank SK for training in the qualitative research.

## Author Contributions

**Conceptualization:** Paisan Ruamviboonsuk, Rajiv Raman.

**Data curation:** Variya Nganthavee, Warisara Pattanapongpaiboon.

**Formal analysis:** Geetha Kumar, Saranya Velu, Raju Sapkota.

**Methodology:** Paisan Ruamviboonsuk, Rajiv Raman.

**Writing – original draft:** Geetha Kumar, Saranya Velu.

**Writing – review & editing:** Shahina Pardhan, Raju Sapkota, Paisan Ruamviboonsuk, Mongkol Tadarati, Peranut Chotcomwongse, Rajiv Raman.

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
