## [Decision Letter · Decision Letter 0]

5 Oct 2022

PONE-D-22-22394Patient, family member, and health care provider perspective on barriers and facilitators to diabetic retinopathy screening in Thailand: A qualitative studyPLOS ONE

Dear Dr. Raman,

Thank you for submitting your manuscript to PLOS ONE. After careful consideration, we feel that it has merit but does not fully meet PLOS ONE’s publication criteria as it currently stands. Therefore, we invite you to submit a revised version of the manuscript that addresses the points raised during the review process.

We look forward to receiving your revised manuscript.

Kind regards,

Chulaporn Limwattananon, Ph.D.

Academic Editor

PLOS ONE

Journal Requirements:

"No"

"This study was supported by Lions Club International Foundation. SFP2050/UND. The authors thank Dr. Robin Driscoll for administration support."

"No"

Reviewers' comments:

Reviewer's Responses to Questions

**Comments to the Author**

1. Is the manuscript technically sound, and do the data support the conclusions?

Reviewer #1: Partly

Reviewer #2: Partly

2. Has the statistical analysis been performed appropriately and rigorously? 

Reviewer #1: No

Reviewer #2: N/A

3. Have the authors made all data underlying the findings in their manuscript fully available?

Reviewer #1: No

Reviewer #2: No

4. Is the manuscript presented in an intelligible fashion and written in standard English?

Reviewer #1: Yes

Reviewer #2: No

5. Review Comments to the Author

Reviewer #1: 1. Qualitative research required qualify person to be the interviewer or conduct the FGD, please submit CVs of the interviewers to ensure scientific soundness of this articles.

2. Could the authors show the numbers of each participants from each hospital to ensure that the data is not mainly come from one hospital which may not reflect the context of the country?

3. This study exclude patients with no light perception which is the appropriate target group to gain the information about the barrier to access to the health care so the author should discuss this issue in the part of discussion.

4. Please provide the detail of how to sampling the individual key informants because the selection bias might be occurred which can reduce the possibility of generalizability of the study.

Reviewer #2: Dear authors,

Thank you so much for an opportunity to review this manuscript. It is a good study exploring barriers of DR screening as the uptake of the service is apparently low. However, I think the results need to be restructured because the current version does not seem to exhibit barriers and factors influencing compliance to the service. My major and specific comments describe below:

Major comments.

1. The current themes do not represent barriers and facilitators of the DR screening. The current results are predominantly related to awareness/knowledge/behaviors of diabetes/care/DR. However, there is no connection between these findings and the service uptake. This needs to explain more how they affect the service use.

2. There is only one theme ‘Barriers to DR screening’ which is directly related to the objective. While the other themes are likely a background information and situation of patients and care. This is why authors need to restructure the results.

3. Authors currently structure the results by views of different stakeholders. Would it be better to combine the same theme into one and explain it from each perspective? For example: ‘the theme ‘care-seeking practice’ is emerged by all stakeholders, can you combine them into one and expand a little bit more why this impact to the screening uptake.

4. When extracting the data, authors may concern barriers related to service delivery such as human resources, finance, information system, policy, etc. to cover all the barriers comprehensively.

5. Quotations supported for each theme are too many, please be selective. Use only ones that outstanding.

6. I do not quite agree by reporting the percentage of participants by gender or by opinion. This is a qualitative study using FGD. Opinions obtained from FGD are ‘Group’s opinion’, not the opinion of each individual. Although there are some qualitative researchers do that but I think it is not appropriate to show how many percent agree/disagree. Instead, authors should demonstrate all the issues raised by all groups either positive or negative.

7. I must say there are many sentences that are not understandable. I think this manuscript also need an extensive English proof reading.

Specific comments.

Abstract. This need to be revised after the manuscript revision completed.

Introduction.

Line 75: How many diabetic patients have been screened for DR recently? This information should be stated.

Methods:

Line 93-98: Please blind the hospital names and describe them by their characters e.g. hospital type, size, location, etc.

Line 113: As the authors use FGD, the instrument is rather ‘a focus group agenda’, not an interview guide.

Recruitment of health care providers is missed, this should be stated.

Line 116-121: I guess here is a list of questions used in the FGD. These sentences are not completed, some verbs are missed. It might be better to illustrate by table.

Line 146-146: Was the coding text performed? As authors set up some categories prior to analyzing the data, where do these categories come from? Was there any theoretical framework used for data analysis?

Line 151: … thereby setting the stage for 151 theme development … I am not sure what this means?

Line 208-209: Please clarify more how ‘changing hospital’ is related to self-ownership. And How does this link to barriers of the screening.

Line 240-241: ‘ophthalmologists are not responsible for any blindness or complications’ … Is this message a routine practice?

Line 261-263: Barriers for attendance for DR screening …. Do the patients also acknowledge these barriers?

Line 265-266: ‘Doctors-led diabetes management groups to discuss and solve the queries raised by patients were considered an important element in the follow-up of patients’ I am not sure what this means?

Line 269-270: ‘Patients thought a day dedicated to DR screening only would be of benefit.’ I am not sure what this means?

For the theme ‘Care-seeking practice…’ I think the better term to use is ‘behavior’ as I think authors intend to explain behaviour of patients how they seek care.

Discussion

Line 458: Knowing the difference between glaucoma and DR is rather a knowledge, not a health literacy.

The discussion points should also mention issues related to service delivery such as human resources, finance, information systems, or screening policy.

Conclusion need to be revised. The current one does not reflect to the objectives.

6. PLOS authors have the option to publish the peer review history of their article (what does this mean?). If published, this will include your full peer review and any attached files.

Reviewer #1: No

Reviewer #2: No

---

## [Author Response · Author response to Decision Letter 0]

17 Dec 2022

PLOS ONE

Journal Requirements:

Response: The style and templates have changes as per suggested.

Response: Have added the supporting information.

"No"

Response: This study was supported by Lions Club International Foundation. SFP2050/UND. The funders had no role in study design, data collection and analysis, decision to publish, or preparation of the manuscript. 

Response: Amended statement has been included in the cover letter.

"This study was supported by Lions Club International Foundation. SFP2050/UND. The authors thank Dr. Robin Driscoll for administration support."

Response: ‘This study was supported by Lions Club International Foundation. SFP2050/UND. The funders had no role in study design, data collection and analysis, decision to publish, or preparation of the manuscript’. 

Response: Amended statement has been now added in the cover letter. Thank you!

Reviewers' comments:

Reviewer's Responses to Questions

Comments to the Author

1. Is the manuscript technically sound, and do the data support the conclusions?

Reviewer #1: Partly

Reviewer #2: Partly

Response: Please see our further responses to reviewer’s comments, below.

2. Has the statistical analysis been performed appropriately and rigorously?

Reviewer #1: No

Reviewer #2: N/A

Response: Please see our further responses to reviewer’s comments, below.

3. Have the authors made all data underlying the findings in their manuscript fully available?

Reviewer #1: No

Reviewer #2: No

Response: All the data are provided in the manuscript. Data such as summary statistics, the data points behind means, medians and variance measures are not relevant given the qualitative nature of this study. ________________________________________

4. Is the manuscript presented in an intelligible fashion and written in standard English?

Reviewer #1: Yes

Reviewer #2: No

Response: We have tried our best to improve the language in the revised manuscript.

5. Review Comments to the Author

Reviewer #1: 1. Qualitative research required qualify person to be the interviewer or conduct the FGD, please submit CVs of the interviewers to ensure scientific soundness of this articles.

Response: Dr.Shuba Kumar (SK) is a social scientist and founder member of Samarth, an NGO engaged in social science research and training. She is trained in epidemiology and social science, has expertise in research methodology, particularly qualitative research methods and questionnaire validation. SK who is trained in qualitative research conducted a three day training workshop on qualitative research methods which was attended by three of the research team (PC, GK and SV). They observed SK conducting interviews and also carried out mock interviews/FGDs using the study guides. Feedback was provided to them to further improve their techniques following which they commenced doing the FGD. The duration of FGDs varied from 45 to 60 minutes. Under the PC supervision the other authors VN, WP and MT conducted the FGD. The CV of PC has been attached along with his workshop participation certificate. 

2. Could the authors show the numbers of each participants from each hospital to ensure that the data is not mainly come from one hospital which may not reflect the context of the country?

Response: Number of participants from each hospital are listed below. 

Participants Hospital 1 (Srisangwal Hospital in Mae Hong Sorn province) Hospital 2 (Pakchongnana Hospital in Nakornrajasima province) Hospital 3 (Wiengsra Crown Prince Hospital in Surat Thani province) Hospital 4 (Rajavithi Hospital in Bangkok)

 Hospital 5 (Sam Roi Yot Hospital in Prachuap Khiri Khan province)

Patients 9 12 5 11 10

Family members 11 8 6 6 10

Health care providers 6 6 7 7 8

3. This study exclude patients with no light perception which is the appropriate target group to gain the information about the barrier to access to the health care so the author should discuss this issue in the part of discussion. 

Response: Thank you. We have discussed this as following in line number 419-423

Although patients with no light perception are the appropriate group to gain information about the barrier, there are obstacles faced by the visually impaired patients due to lack of nonverbal signals such as hand gestures, nods, eye contacts and facial expression in the FGD. A study by Qui et al. reported having unpleasant feelings during the conversation as a result of not being able to read nonverbal cues such hand gestures, nods, eye contact, and facial expressions during face-to-face communication. To completely understand the challenges experienced in receiving diabetic care, future studies must concentrate on individuals who are visually impaired.

4. Please provide the detail of how to sampling the individual key informants because the selection bias might be occurred which can reduce the possibility of generalizability of the study.

Response: The selection of patients was done purposively using maximum variation sampling. We looked at the retinal specialists' appointment schedules with a focus on the obstacles to DR care. We would visit the outpatient department of the particular retinal expert on the day of the appointment as scheduled. We recruited men and women aged 18 years or more with diabetes at least for 5 years or more belonging to different age, education and occupation. Selection of a varied group of participants enabled us to obtain diverse perceptions on the study topic which was to understand, perceptions and awareness about DR, the need for regular eye screening, perceived barriers faced in seeking eye care, in addition to awareness about diabetes and diabetic related complications. Being a qualitative study, generalizations are not possible as the samples are typically small and purposively selected but they enable deeper insights and understanding. This has been now discussed in line number 110-113. 

Reviewer #2: Dear authors,

Thank you so much for an opportunity to review this manuscript. It is a good study exploring barriers of DR screening as the uptake of the service is apparently low. However, I think the results need to be restructured because the current version does not seem to exhibit barriers and factors influencing compliance to the service. My major and specific comments describe below:

Major comments.

1. The current themes do not represent barriers and facilitators of the DR screening. The current results are predominantly related to awareness/knowledge/behaviors of diabetes/care/DR. However, there is no connection between these findings and the service uptake. This needs to explain more how they affect the service use.

Response: Thank you for the suggestion. We have now included the following section explaining the connection between our findings and the service uptake in Page number 8-17

2. There is only one theme ‘Barriers to DR screening’ which is directly related to the objective. While the other themes are likely a background information and situation of patients and care. This is why authors need to restructure the results.

Response: We have now reconstructed the result part. 

3. Authors currently structure the results by views of different stakeholders. Would it be better to combine the same theme into one and explain it from each perspective? For example: ‘the theme ‘care-seeking practice’ is emerged by all stakeholders, can you combine them into one and expand a little bit more why this impact to the screening uptake.

Response: Thank you for this comment to modify. We have now re-structured the result as following to include your suggestion in page number 10-11.

4. When extracting the data, authors may concern barriers related to service delivery such as human resources, finance, information system, policy, etc. to cover all the barriers comprehensively.

Response: Thank you. These have been now discussed as following in Page number 14-15.

Barriers to DR screening included a lack of availability of ophthalmologists, long waiting times, financial constraints, transportation problems, the absence of accompanying individuals to attend appointment, and ignorance of DR screening. These challenges were cited by patients as reasons for not going to the hospital. Other reasons including long travel time and forgetting the appointment date. 

5. Quotations supported for each theme are too many, please be selective. Use only ones that outstanding.

Response: Selective quotes are now added.

6. I do not quite agree by reporting the percentage of participants by gender or by opinion. This is a qualitative study using FGD. Opinions obtained from FGD are ‘Group’s opinion’, not the opinion of each individual. Although there are some qualitative researchers do that but I think it is not appropriate to show how many percent agree/disagree. Instead, authors should demonstrate all the issues raised by all groups either positive or negative.

Response: We agree with your suggestion and we have modified the contents in the result part accordingly.

7. I must say there are many sentences that are not understandable. I think this manuscript also need an extensive English proof reading.

Response: We have tried our best to improve English and simplify the sentences in the revised manuscript.

Specific comments.

Abstract. This need to be revised after the manuscript revision completed.

Response: This has been updated as per the revisions.

Introduction.

Line 75: How many diabetic patients have been screened for DR recently? This information should be stated.

Response: This has been stated in line number 74-75. 

Methods:

Line 93-98: Please blind the hospital names and describe them by their characters e.g. hospital type, size, location, etc.

Response: Thank you for the suggestion. We have modified this to address your comment. (line number 97-99)

Line 113: As the authors use FGD, the instrument is rather ‘a focus group agenda’, not an interview guide.

Response: Thank you! We have revised the manuscript as you suggested (line number 115) 

Comment: Recruitment of health care providers is missed, this should be stated.

Response: This has been now stated as following in the revised manuscript (line number 116-118)

Line 116-121: I guess here is a list of questions used in the FGD. These sentences are not completed, some verbs are missed. It might be better to illustrate by table.

Response: Have re-framed the lines now. (119-123)

Line 146-146: Was the coding text performed? As authors set up some categories prior to analyzing the data, where do these categories come from? Was there any theoretical framework used for data analysis?

Response: We did use the framework analytical approach which is also termed as thematic analysis or content analysis. Furthermore, our research included young researchers who were not highly experienced in qualitative data analysis. The framework approach which involves a structured and systematic method of organising our data into spreadsheets was something they could understand and undertake. (Line number 143-148)

Line 151: … thereby setting the stage for 151 theme development … I am not sure what this means?

Response: This was a typo. This sentence has been revised as following to add clarity line number (151-153)

Line 208-209: Please clarify more how ‘changing hospital’ is related to self-ownership. And How does this link to barriers of the screening.

Response: Generally when the glucose level is not get controlled patients may have a tendency to visit another hospital for better care. Have rephrased the line. (line number:181-183)

Line 240-241: ‘ophthalmologists are not responsible for any blindness or complications’ … Is this message a routine practice?

Response: It is not used in a routine practice, but the message focuses on the risks that were raised, such as the type and effects of the treatment, the results of the surgery, or the patient concerns around the disease prognosis. The line has rephrased now in the line number 252-257.

Line 261-263: Barriers for attendance for DR screening …. Do the patients also acknowledge these barriers?

Response: Yes, these were the barriers reported by the patient during the FGD (line number:302-306) 

Line 265-266: ‘Doctors-led diabetes management groups to discuss and solve the queries raised by patients were considered an important element in the follow-up of patients’ I am not sure what this means?

Response: This sentence has been revised as following in line number 338-339

‘People preferred a discussion group with doctors to address their concerns and questions about managing their diabetes and encourage compliance.’ 

Line 269-270: ‘Patients thought a day dedicated to DR screening only would be of benefit.’ I am not sure what this means?

Response: This sentence has been revised as following to add clarity in line 341-342

For the theme ‘Care-seeking practice…’ I think the better term to use is ‘behavior’ as I think authors intend to explain behaviour of patients how they seek care.

Response: Thank you! We have renamed the theme as per your suggestion (Page number 10)

Discussion

Line 458: Knowing the difference between glaucoma and DR is rather a knowledge, not a health literacy.

Response: Thank you! We have revised this as per your suggestion (line number 402-403)

The discussion points should also mention issues related to service delivery such as human resources, finance, information systems, or screening policy.

Response: These points have been now discussed in line number 436-438 in the revised manuscript. 

Conclusion need to be revised. The current one does not reflect to the objectives.

Response: We have revised the conclusion as following to reflect to the objectives. Line number 453-457.

‘The majority of patients were passive and had great faith in the HCPs. Family participation in DM care can improve health-related behaviors in patients. The involvement of non-physician providers working in the field of diabetes has a significant impact in encouraging patients to attend DR screening. Our findings may be useful in designing policies to make DR screening in Thailand more accessible’

6. PLOS authors have the option to publish the peer review history of their article (what does this mean?). If published, this will include your full peer review and any attached files.

Do you want your identity to be public for this peer review? For information about this choice, including consent withdrawal, please see our Privacy Policy.

Reviewer #1: No

Reviewer #2: No

---

## [Decision Letter · Decision Letter 1]

24 Mar 2023

PONE-D-22-22394R1Patient, Family Member, and Health Care Provider Perspective on Barriers and Facilitators to Diabetic Retinopathy Screening in Thailand: A Qualitative StudyPLOS ONE

Dear Dr.,

Thank you for submitting your manuscript to PLOS ONE. After careful consideration, we feel that it has merit but does not fully meet PLOS ONE’s publication criteria as it currently stands. Therefore, we invite you to submit a revised version of the manuscript that addresses the points raised during the review process.

We look forward to receiving your revised manuscript.

Kind regards,

Pracheth Raghuveer, MD, DNB

Academic Editor

PLOS ONE

Journal Requirements:

Reviewers' comments:

Reviewer's Responses to Questions

**Comments to the Author**

1. If the authors have adequately addressed your comments raised in a previous round of review and you feel that this manuscript is now acceptable for publication, you may indicate that here to bypass the “Comments to the Author” section, enter your conflict of interest statement in the “Confidential to Editor” section, and submit your "Accept" recommendation.

Reviewer #1: All comments have been addressed

Reviewer #2: All comments have been addressed

2. Is the manuscript technically sound, and do the data support the conclusions?

Reviewer #1: Yes

Reviewer #2: Partly

3. Has the statistical analysis been performed appropriately and rigorously? 

Reviewer #1: N/A

Reviewer #2: N/A

4. Have the authors made all data underlying the findings in their manuscript fully available?

Reviewer #1: Yes

Reviewer #2: No

5. Is the manuscript presented in an intelligible fashion and written in standard English?

Reviewer #1: Yes

Reviewer #2: No

6. Review Comments to the Author

Reviewer #1: (No Response)

Reviewer #2: Dear authors

Thank you for the revised manuscript which has been improved significantly. However, I still have two points which I think it may not be the case:

1. Discussion and conclusion about a need of non-physician clinician involvement, as I do not see this finding mentioned in the result section yet. So, this point of discussion and conclusion is not aligned to the findings. Please check and revise.

2. The point regarding equity of health insurance scheme (line 372-375, 440-441) since the authors have stated in line 323 “Despite screening service for diabetes being free...” which I think the DR screening is included in the routine service for diabetic patients and covered all insurance scheme. Is this correct? If so, then this point may not be the case. Please check and revise.

Abstract

Line 51-52: is this matched to the findings described in maintext?

Line 55-59: Same as point 1 & 2 above.

Grammatic errors and other little things

Line 319: … did not adopted …

Line 345: … that that they could ….

Line 346: … We have a Line group …. An explanation in brackets of “Line” is needed e.g. a chat application popularly used in Thailand or something like this.

Kind regards

7. PLOS authors have the option to publish the peer review history of their article (what does this mean?). If published, this will include your full peer review and any attached files.

Reviewer #1: No

Reviewer #2: No

---

## [Author Response · Author response to Decision Letter 1]

31 Mar 2023

Journal Requirements:

Response: We have reviewed the reference list as per suggested. 

Reviewers' comments:

Reviewer's Responses to Questions

Comments to the Author

1. If the authors have adequately addressed your comments raised in a previous round of review and you feel that this manuscript is now acceptable for publication, you may indicate that here to bypass the “Comments to the Author” section, enter your conflict of interest statement in the “Confidential to Editor” section, and submit your "Accept" recommendation.

Reviewer #1: All comments have been addressed

Reviewer #2: All comments have been addressed

Response: Thank you for your review and insightful comments. We found them very useful in improving our manuscript.

2. Is the manuscript technically sound, and do the data support the conclusions?

Reviewer #1: Yes

Reviewer #2: Partly

Response: We have further revised the manuscript to make it technically sound better, wherein data directly support the conclusion. Please see our ‘further responses to reviewer’s comments’, below.

3. Has the statistical analysis been performed appropriately and rigorously?

Reviewer #1: N/A

Reviewer #2: N/A

Response: Thank you.

4. Have the authors made all data underlying the findings in their manuscript fully available?

Reviewer #1: Yes

Reviewer #2: No

Response: All the data are provided in the results and the supplementary materials. The findings will be also deposited in Anglia Ruskin Research Online (ARRO). We have tried our best to cover all the findings in the revised manuscript.

5. Is the manuscript presented in an intelligible fashion and written in standard English?

Reviewer #1: Yes

Reviewer #2: No

Response: We have fixed the typos and tried our best to improve the language in the revised manuscript.

6. Review Comments to the Author

Reviewer #1: (No Response)

Reviewer #2: Dear authors

Thank you for the revised manuscript which has been improved significantly. However, I still have two points which I think it may not be the case:

1. Discussion and conclusion about a need of non-physician clinician involvement, as I do not see this finding mentioned in the result section yet. So, this point of discussion and conclusion is not aligned to the findings. Please check and revise.

Response: Thank you for highlighting this important point. Apologies for the missing point, we have included non-physician clinician involvement throughout the ‘Results’ section. 

‘Family members and, non-physician clinicians (such as diabetes nurses, diabetes educators, physician assistants) who work in the field of diabetes play a vital role in encouraging patients to attend diabetes and DR follow-ups visits regularly (Abstract, page 3). 

‘Therefore physician and, non-physician clinicians felt that patients should be given a guidebook, posters, books, videos, or brochures with the necessary information reminding them about the possible complications of uncontrolled diabetes in eyes, feet, etc. (Page 11, last paragraph). 

‘Patients who are unable to travel to the hospital due to distance, financial constraints, or other factors should be screened at a primary care center near their home with the help of non-physician clinicians.’ (Page 16, last paragraph). 

‘One caregiver identified social stigma, for example, poorer patients felt ignored by the hospital, and suggested that education should be provided by non-physician clinicians (nurse/nurse aid) to poor and elderly patients.’(Page 16, last paragraph)

‘The involvement of non-physician clinicians working in the field of diabetes has a significant impact on encouraging patients to attend DR screening (Page 20, Conclusion section).

2. The point regarding equity of health insurance scheme (line 372-375, 440-441) since the authors have stated in line 323 “Despite screening service for diabetes being free...” which I think the DR screening is included in the routine service for diabetic patients and covered all insurance scheme. Is this correct? If so, then this point may not be the case. Please check and revise.

Response:. Thank you. The point have been re-phrased as “Despite screening service for diabetes being covered by insurance scheme” in the revised manuscript. (Page number 15, line number 320)

Abstract

Line 51-52: is this matched to the findings described in maintext?

Response: Thank you we have added the relevant text now (Line number 51-52) 

Line 55-59: Same as point 1 & 2 above.

Response: We have re-phrased the line now (Line number 55-57)

Grammatic errors and other little things

Line 319: … did not adopted …

Response: Thank you! This was a typo and has been now corrected. (Line number 390)

Line 345: … that that they could ….

Response: Thank you! This was a typo and has been now corrected. (Line number 342)

Line 346: … We have a Line group …. An explanation in brackets of “Line” is needed e.g. a chat application popularly used in Thailand or something like this.

Kind regards

Response: Line group is a freeware application in iOS, android and window software. Instant communication by text, photographs, and voice can be created. The line group comprising many people can be used to communicate in a group. Only invitees can enter to line group. This has been mentioned in the revised manuscript in Page 16, line number 344-345.

‘“We have a “Line group” (freeware application software to communicate with many patients) for diabetic patients as well, in which we can change our appointment date within 7 days and also consult for the medications. (FGD 5)”

7. PLOS authors have the option to publish the peer review history of their article (what does this mean?). If published, this will include your full peer review and any attached files.

Do you want your identity to be public for this peer review? For information about this choice, including consent withdrawal, please see our Privacy Policy.

Reviewer #1: No

Reviewer #2: No

Response: Thank you.

---

## [Decision Letter · Decision Letter 2]

5 Jun 2023

PONE-D-22-22394R2Patient, Family Member, and Health Care Provider Perspective on Barriers and Facilitators to Diabetic Retinopathy Screening in Thailand: A Qualitative StudyPLOS ONE

Dear Dr. Rajiv Raman

Thank you for submitting your manuscript to PLOS ONE. After careful consideration, we feel that it has merit but does not fully meet PLOS ONE’s publication criteria as it currently stands. Therefore, we invite you to submit a revised version of the manuscript that addresses the points raised during the review process.

We look forward to receiving your revised manuscript.

Kind regards,

Pracheth Raghuveer, MD, DNB

Academic Editor

PLOS ONE

Journal Requirements:

Reviewers' comments:

Reviewer's Responses to Questions

**Comments to the Author**

1. If the authors have adequately addressed your comments raised in a previous round of review and you feel that this manuscript is now acceptable for publication, you may indicate that here to bypass the “Comments to the Author” section, enter your conflict of interest statement in the “Confidential to Editor” section, and submit your "Accept" recommendation.

Reviewer #1: All comments have been addressed

2. Is the manuscript technically sound, and do the data support the conclusions?

Reviewer #1: Yes

3. Has the statistical analysis been performed appropriately and rigorously? 

Reviewer #1: Yes

4. Have the authors made all data underlying the findings in their manuscript fully available?

Reviewer #1: Yes

5. Is the manuscript presented in an intelligible fashion and written in standard English?

Reviewer #1: Yes

6. Review Comments to the Author

Reviewer #1: please rephrase the sentence in the limitation of the study p20 line 448; because the reason of limited generalizability is no sampling method for choosing the areas of interest to cover all areas in the country which might have different context in healthcare service, believe and socioeconomic status.

7. PLOS authors have the option to publish the peer review history of their article (what does this mean?). If published, this will include your full peer review and any attached files.

Reviewer #1: No

---

## [Author Response · Author response to Decision Letter 2]

9 Jun 2023

Journal Requirements:

Response: We confirm this.

Reviewers' comments:

Reviewer's Responses to Questions

Comments to the Author

1. If the authors have adequately addressed your comments raised in a previous round of review and you feel that this manuscript is now acceptable for publication, you may indicate that here to bypass the “Comments to the Author” section, enter your conflict of interest statement in the “Confidential to Editor” section, and submit your "Accept" recommendation.

Reviewer #1: All comments have been addressed

Response: Thank you

2. Is the manuscript technically sound, and do the data support the conclusions?

Reviewer #1: Yes

Response: Thank you 

3. Has the statistical analysis been performed appropriately and rigorously?

Reviewer #1: Yes

Response: Thank you. 

4. Have the authors made all data underlying the findings in their manuscript fully available?

Reviewer #1: Yes

Response: Thank you

5. Is the manuscript presented in an intelligible fashion and written in standard English?

Reviewer #1: Yes

Response: Thank you 

6. Review Comments to the Author

Reviewer #1: please rephrase the sentence in the limitation of the study p20 line 448; because the reason of limited generalizability is no sampling method for choosing the areas of interest to cover all areas in the country which might have different context in healthcare service, believe and socioeconomic status.

Response: Thank you for the suggestions. We have re-phrased the sentence (P20 line 448) in the revised manuscript as following. 

‘Due to the lack of a sampling strategy that guarantees coverage of every region in Thailand, the findings may not be generalizable across the whole country due to potential regional differences in socioeconomic position, healthcare services, and beliefs. 

7. PLOS authors have the option to publish the peer review history of their article (what does this mean?). If published, this will include your full peer review and any attached files.

Do you want your identity to be public for this peer review? For information about this choice, including consent withdrawal, please see our Privacy Policy.

Reviewer #1: No

---

## [Editor Report · Decision Letter 3]

24 Jul 2023

Patient, Family Member, and Health Care Provider Perspective on Barriers and Facilitators to Diabetic Retinopathy Screening in Thailand: A Qualitative Study

PONE-D-22-22394R3

Dear Dr. Rajiv Raman,

We’re pleased to inform you that your manuscript has been judged scientifically suitable for publication and will be formally accepted for publication once it meets all outstanding technical requirements.

Kind regards,

Pracheth Raghuveer, MD, DNB

Academic Editor

PLOS ONE
---

## [Editor Report · Acceptance letter]

26 Jul 2023

PONE-D-22-22394R3 

Patient, Family Member, and Health Care Provider Perspective on Barriers and Facilitators to Diabetic Retinopathy Screening in Thailand: A Qualitative Study 

Dear Dr. Raman:

I'm pleased to inform you that your manuscript has been deemed suitable for publication in PLOS ONE. Congratulations! Your manuscript is now with our production department. 

Kind regards, 

on behalf of

Dr. Pracheth Raghuveer 

Academic Editor

PLOS ONE